# Effects of New Complementary Feeding Guidelines on Eating Behaviour, Food Consumption and Growth in Colombian Children: 6-Year Follow-Up of a Randomised Controlled Trial

**DOI:** 10.3390/nu16142311

**Published:** 2024-07-18

**Authors:** Gilma Olaya Vega, Mary Fewtrell

**Affiliations:** 1Nutrition and Biochemistry Department, Faculty of Sciences, Pontificia Universidad Javeriana, Bogota 110231, Colombia; 2Population, Policy & Practice Research and Department, UCL GOS Institute of Child Health, University College London (UCL), London WC1N 1EH, UK; m.fewtrell@ucl.ac.uk

**Keywords:** complementary feeding, effects, eating behaviour, growth, infants, children

## Abstract

Complementary feeding (CF) may influence later eating behaviour and growth. Our previous Randomised Control Trial (RCT) reported that new CF guidelines (NCFGs) implemented in 6–12-month-old infants in Bogota, Colombia, had positive short-term effects on red meat, vegetable and fruit consumption. Here, we assessed the effects of the NCFGs on food consumption, eating behaviour and growth at 6 years of age. Weight and height were measured using 50 children (58.8%) from the cohort. Feeding behaviour was measured using the Child Eating Behaviour Questionnaire (CEBQ) and maternal and child food consumption was measured using a semi-quantitative food frequency questionnaire. The control group (CG) had a significantly higher weekly consumption of chocolate milk drinks (*p* = 0.05). The mean food responsiveness (FR) score was significantly higher in the CG (*p* < 0.001). Although HAZ (height for age Z-score) at 6 years of age was significantly higher in the CG (*p* < 0.02), there was no significant difference between groups in the change in HAZ from 6 months and 12 months to 6 years of age. BMIZ (body mass index Z-score) and % overweight (CG 18.5% versus NCFG 13%) or obese (3.7% versus 0%) were not significantly different between groups. BMIZ was positively predicted by FR (β 0.293; *p* = 0.014) and negatively predicted by weekly red meat consumption episodes per week at 12 months (β −0.169; *p* = 0.020). Although there was no direct effect of an intervention on BMIZ at 6 years of age, the results were consistent with an indirect effect via intervention effects on meat consumption at an age of 12 months and FR at 6 years of age. However, further longitudinal studies with a larger sample size are needed.

## 1. Introduction

The mode of feeding during the first year of life has both short- and longer-term effects on growth, health and nutritional status and is considered to be a window of opportunity to improve these outcomes. This period is also important to promote positive food habits and eating behaviour that influence food preferences, which are related to the risk of obesity during childhood and later in life [1,2,3]. Complementary feeding (CF) is often introduced too early with inadequate or inappropriate food, which can lead to micronutrient deficiencies, infectious diseases and growth failure [4,5,6]. The World Health Organization (WHO) recommends exclusive breastfeeding for 6 months, continued alongside complementary feeding until two years of age [7]. After 6 months of age, complementary feeding has to fill the gap for important nutrients, especially protein, iron and zinc; however, some studies have reported a relationship between a high protein intake during this period and higher BMI and body fat in later life [1,8,9].

In recent years, the prevalence of childhood obesity has increased worldwide in children under 5 years of age, with more than 42 million children affected [10]. This may reflect an increase in high-energy-density food consumption, including sugar-sweetened drinks [11,12], together with a shift in physical activity and feeding behaviour. This may have long-term consequences as overweight and obese children are more likely to be obese in adult life. However, there are limited studies exploring the effect of complementary feeding practices on food consumption, eating behaviour and growth in mid-childhood.

In this study, we investigated the effects of a complementary feeding intervention on growth, food consumption and eating behaviour at 6 years of age. We studied a cohort of children living under socioeconomic constraints in Bogota, Colombia, who participated in an RCT of new complementary feeding guidelines (NCFGs) based on the following three main messages that were emphasised at all study visits between 6 and 12 months of age: (i) the importance of continuing breastfeeding alongside CF; (ii) the importance of including red meat at least 3 times per week as a source of iron to prevent anaemia; and (iii) the importance of daily fruit and vegetable consumption as part of a healthy diet. At 12 months of age, compared with the control group (CG), infants in the NCFG group were more likely to have daily consumption of red meat, fruit and vegetables, the frequency of red meat consumption per week was significantly greater and changes in haemoglobin and haematocrit from 6 to 12 months were significantly higher, while changes in LAZ and WLZ from 6 to 12 months were not significantly different between groups [13]. We hypothesised that the NCFG intervention would have a medium-term effect on food consumption, eating behaviour and growth at 6 years of age.

## 2. Methods

### 2.1. Subjects and Study Design

#### 2.1.1. Original Trial

Healthy term infants with a birth weight > 2500 g participated in an RCT (ISRCTN57733004) of standard/current feeding guidelines (Control group (CG)) versus new complementary feeding guidelines (NCFGs) starting at 6 months of age, with follow-ups up to 12 months of age and who were living under socioeconomic constraints in Bogota, Colombia [13]. A low socioeconomic status was defined as a socioeconomic stratification level of 1–3 out of 6. In these families, income is mainly derived from informal and part-time jobs, most live in rented or shared households and approximately 29% of this population experiences food insecurity, although 97% of the population has public service coverage, including tap water, electricity and waste disposal [13]. All mothers at two hospitals in Bogota (Suba and Fontibon) were approached when their infant was 4 months old and were advised to exclusively breastfeed (EBF), then continue with breast milk (BM) alongside CF. Infants who met the inclusion criteria were randomly assigned to either the NCFG group or the current complementary feeding practice control group (CG). In total, 85 infants were randomised (42 to the NCFG group and 43 to the CG). The random assignment was stratified by hospital and whether CF had been introduced to infants between 4 and 6 months of age. Infants randomly assigned to the NCFG group received individual nutrition counselling as previously described, with face-to-face sessions and detailed verbal and written guidance (leaflets) that included suitable complementary food, recipes, food preparation, the amount of food that should be offered and food hygiene. The acceptability, tolerance and affordability of recommendations in the NCFG group were assessed by a questionnaire completed by mothers when infants were 8, 10 and 12 months of age [13].

#### 2.1.2. The 6-Year Follow-Up

The follow-up study was approved by the Research Ethical Committee at the Pontificia Universidad Javeriana, Bogota, Colombia. Children eligible for this study were participants in the RCT of the NCFGs who were contacted by a telephone call. Parents of children from the original trial that agreed to participate gave written informed consent before data collection in the clinic (Suba Hospital) where the study took place.

### 2.2. Anthropometric Measurements at 6 Years of Age

Weight was measured using a scale (SECA 874) with precision of 50 g and height was measured using a stadiometer (SECA 213) with precision of 1 mm. All measurements were performed following a protocol and the mean of 3 measurements was used. BMI was calculated as the weight in kilograms divided by the square of the height in metres (kg/m^2^). The results were converted to indicators standardised for age and sex (BMI, BMIZ and height (HAZ)) using the World Health Organization reference cut-off points to define being underweight (BMI < −2 SD), overweight (BMI > +1 SD) and obese (BMI > +2 SD) [14]. The weight and stature of the mothers were measured at the clinic visit and data were converted into BMI.

### 2.3. Assessment of Frequency of Food Intake

A semi-quantitative food frequency questionnaire (FFQ) adapted for the food available in Colombia was used to assess the frequency (daily, weekly and monthly) of the consumption of milk, milk dairy products, fruit, vegetables, meat, eggs, cereals, tubers, soft drinks, sugar and processed food. This was the same FFQ (validated against a 24 h recall in an unpublished pilot study) used in the original trial at 12 months and was applied separately for children and mothers [13].

### 2.4. Assessment of Eating Behaviour

The child’s eating behaviour was assessed using the CEBQ translated into Spanish, previously tested for comprehension in a pilot study using preschool children (3 to 5 years) that measured food habits and feeding behaviour (unpublished data). The CEBQ consists of the following eight scales [15]: food responsiveness (FR), enjoyment of food (EF), desire to drink (DD), satiety responsiveness (SR), slowness in eating (SE), food fussiness (FF), emotional overeating (EOE) and emotional undereating (EUE). There are 35 items in total. To assess the intensity of the behaviour, we used a Likert-type scale that included an option for parents to score their child’s eating behaviour on a scale of 1–5 (never = 1, rarely = 2, sometimes = 3, often = 4 and always = 5) for each item. Some subscales such as FR and EF assess behaviour related to a higher food intake and others, such as SR and SE, assess the opposite behaviour.

## 3. Statistics

Data analyses were performed using SPSS version 26. The mean and standard deviation (SD) for continuous variables or proportions (%) were used to describe the characteristics. Outcomes were compared between randomised groups using a *t*-test for normally distributed variables, a Mann–Whitney test for variables with a non-normal distribution and a chi-squared test for categorical variables. Simple correlations and multiple regression analyses were also used to investigate the relationship between the mode of feeding in the first year of life and childhood variables (food consumption, eating behaviour and growth) at 6 years of age. Z-scores for anthropometric indicators such as BMIZ and HAZ were used as the main outcomes in the analyses. We conducted stepwise regression analyses to investigate the predictors of BMIZ and HAZ at 6 years of age. We first investigated the univariate associations between the baseline variables (birth weight, child’s sex, mother’s educational level, mother’s occupation, mother’s BMI and intervention group (GC-NCFG)) and post-randomisation variables (FR and weekly red meat consumption at 12 months as number of times) and BMIZ at 6 years of age. The HAZ baseline variables were the mother’s height, birth length, sex, number of children, mother’s educational level, mother’s occupation for HAZ at 6 months, the intervention group (ie. CG or NCFG) and post-randomisation WLZ at 12 months. Variables with a significant association were then included in the regression models with BMIZ or HAZ as the dependent variable. Model 1 included the baseline variables plus the intervention group and model 2 included the baseline variables and the intervention group along with the post-randomisation variables.

## 4. Results

The eligible sample comprised 85 healthy term infants who had participated in the original RCT, but contact was lost with 25 families. The remaining 60 families were contacted and invited to participate in the follow-up study. Of those contacted, 10 mothers did not agree to participate because they did not have enough time to visit the clinic for the data collection. Therefore, 50 children were included (Figure 1). Attrition (58.8%) mainly resulted from mothers changing their address and telephone number, whilst others moved away from Bogota. There were no significant differences in the baseline characteristics for the 50 infants that were seen at 6 years of age versus the 35 infants who were not seen (Table 1). At 6 years of age, there was no significant difference between the NCFG group and CG for sex, EBF duration, total breastfeeding, age of complementary feeding and mother’s age or stature, but children in the CG had significantly higher LAZ at birth and WLZ at 12 months of age (Table 2) than infants in the NCFG group.

### 4.1. Effects of NCFGs on Growth at 6 Years of Age

At 6 years of age, there was no significant difference between the NCFG group and CG in BMI and BMIZ, nor was there a significant change in BMIZ between 6 months or 12 months and 6 years of age. HAZ at 6 years of age was significantly higher in the CG, although there was no significant difference in the change in HAZ from 6 or 12 months to 6 years of age (Table 3). The proportion of children who were overweight was lower in the NCFG group than in the CG (13% versus 18.5%; not significant). One child in the CG was obese versus none in the NCFG.

### 4.2. Food Intake

There was no significant difference between children’s weekly consumption of milk (10.2 (±4.3) versus 8.1 (±5.0)), fruit (14.9 (±5.3) versus 13.5 (±5.4)), red meat (3.5 (±1.8) versus 3.5 (±2.0)) or vegetables (8.7 (±4.7) versus 8.1 (±4.2)) between the NCFG group and CG, although there was a trend for more frequent milk, fruit and vegetable consumption and also a trend towards a more frequent soft drink consumption in the NCFG group (Table 3). However, CG children had a significantly more frequent consumption of chocolate milk drinks (5.9 (±5.1) versus 3.5 (±2.4); *p* < 0.05) (Table 3).

The comparison of weekly food consumption between mothers in the NCFG group and CG showed a significantly lower consumption of soft drinks (*p* < 0.05), sugary drinks (*p* < 0.05) and processed food (*p* < 0.05) (Table 3) in the NCFG group. There was a non-significant trend for higher weekly red meat consumption in the NCFG group (4.6 (±3.6) versus 3.3 (±2.1)) as well as vegetables (12.0 (±9.8) versus 8.0 (±6.2)); milk (7.45 (±6.4) versus 7.38 (±5.8)) and fruit (11.8 (±6.0) versus 11.6 (±14.6)).

Positive correlations were found between the frequency of food consumption of children and mothers (Table 4). The daily and weekly milk consumption of children positively correlated with daily (r = 0.539; *p* = 0.0001) and weekly (r = 0.341; *p* = 0.029) maternal milk consumption as well as with daily (r = 0.417; *p* = 0.007) and weekly maternal dairy product consumption (r = 0.413; *p* = 0.007). Similar correlations were found for daily (r = 0.339; *p* = 0.030) and weekly red meat consumption (r = 0.599; *p* = 0.0001), daily (r = 0.528; *p* = 0.0001) and weekly vegetable consumption (r = 0.465; *p* = 0.003) and daily (r = 0.398; *p* = 0.011) and weekly fruit consumption (r = 0.408; *p* = 0.009) as well as red meat consumption ≥ 3 times per week (r = 0.437; *p* = 0.004), consumption per week of soft drinks (r = 0.382; *p* = 0.015) and weekly consumption of sweets (r = 0.434; *p* = 0.005).

### 4.3. Eating Behaviour

Children from the NCFG group had a significantly lower FR score (2.45 (±1.1)) compared with those in the CG (3.4 (±0.9); *p* < 0.001) (Table 3), with no significant difference observed for the other CEBQ measures.

### 4.4. Predictors of BMIZ at 6 Years of Age

In the univariate analyses, the only baseline variable significantly correlated with BMIZ at 6 years of age was the mother’s BMI (r = 0.346; *p* = 0.025), while the post-randomisation variables significantly correlated with BMIZ at 6 years of age were FR (r = 0.312; *p* = 0.027) and weekly red meat consumption at 12 months (r = −0.427; *p* = 0.002). In model 1, including the mother’s BMI and randomised group, BMIZ at 6 years of age was predicted by the mother’s BMI as β 0.084 (95% CI 0.011 to 0.158, *p* = 0.025). In model 2, with the addition of post-randomisation variables, BMIZ at 6 years of age was predicted by FR (β 0.293 (95% CI 0.064 to 0.523); *p* = 0.014) and weekly red meat consumption at 12 months (β −0.169 (95% CI −0.310 to −0.028); *p* = 0.020) (Table 5).

### 4.5. Predictors of HAZ at 6 Years of Age

In the univariate analyses, HAZ at 6 years of age was associated with mother’s height (r = 0.454; *p* = 0.003), LAZ at 6 months (r = 0.726; *p* = 0.0001), randomised group (CG-NCFG) (r = −0.328; *p* = 0.020) and LAZ at 12 months (r = 0.625; *p* = 0.0001). In model 1, HAZ at 6 years of age was predicted by HAZ at 6 months (β 0.656 (95% CI 0.468 to 0.844; *p* = 0.0001)). In model 2, with the addition of LAZ at 12 months, the only significant predictor was also LAZ at 6 months (β 0.683 (95% CI 0.491 to 0.874; *p* = 0.0001)) (Table 5).

## 5. Discussion

In this study, we tested the hypothesis that the NCFG intervention that promoted red meat consumption at least three times per week as well as daily fruit and vegetable consumption and continued breastfeeding alongside complementary feeding from 6 to 12 months would have longer-term effects on food consumption, eating behaviour and growth at 6 years of age.

At 12 months, infants in the NCFG group had a significantly higher consumption of red meat, fruit and vegetables versus the CG. These effects were not maintained at 6 years of age, although there were trends towards more frequent meat, fruit and vegetable consumption in the NCFG group, whereas the CG had a significantly more frequent consumption of chocolate milk drinks. The lack of significant differences in vegetable, fruit and meat consumption between randomised groups may have been due to the children’s exposure to school feeding programs designed to improve food access and food habits in children from families living in poor socioeconomic conditions; these programs provide breakfast, lunch and snacks as part of the child’s school schedule.

Mothers in the NCFG group had a trend towards a more frequent daily consumption of red meat, fruit and vegetables and child food frequency consumption was positively correlated with maternal consumption of milk, milk dairy products, vegetables, fruit and red meat. These findings suggest that the child’s food consumption is influenced by the mother’s decisions, as has been previously reported [16,17,18]. It is possible that maternal choices in terms of food purchasing may have been influenced by the nutritional messages promoted in the NCFGs. Mothers in the NCFG group also had a significantly less frequent consumption of soft drinks and sugary drinks, although there was no correlation between the mother’s and child’s consumption.

FR is defined as the child’s drive to eat [15]. It has also been described as “the … level of feeding demandingness, responsiveness to milk and feeding cues, and propensity to eat more than provided” [19]. FR is thought to be determined both genetically and by environmental factors such as infant feeding and it may be associated with weight gain and obesity risk [19,20,21]. The mean FR score was significantly higher in the CG (*p* < 0.002) at 6 years of age, suggesting that the NCFG recommendations of continued breastfeeding alongside CF from 6 to 12 months influenced FR. This may have occurred as a result of the mother’s or caregiver’s behaviour in terms of feeding practices and food choices. This is consistent with previous studies suggesting that EBF plus adequate CF practices are likely to influence a child’s future eating behaviour, feeding practices and self-regulation of food intake [18,22,23]. A previous study [21] showed that parental behaviour, such as pressuring the infant to eat, significantly increased FR from 2 to 6 months (*p* = 0.004); during complementary feeding, the pressure to eat cereal from 6 to 14 months was a predictor of increased FR (*p* = 0.02). In a literature review, it was reported that CF using the baby-led weaning technique was associated with lower FR [24], but there is a need for further studies to investigate the influence of CF practices and nutrient intake on appetite, self-regulation and FR in the mid- and long term.

Although this study was not powered to detect differences in being overweight or obese, there was a non-significant trend with a higher proportion of children being overweight in the CG. In the multivariate models examining the predictors of BMIZ at 6 years of age, we found that BMIZ was positively predicted by FR and negatively predicted by weekly red meat consumption at 12 months, although there was no independent effect of the intervention. This was consistent with a study [25] of children who were 7 to 12 years old, which reported that FR was positively associated with weight and the predicted BMI Z-score. On the other hand, another study [26] found a positive correlation between FR and BMI at 6 years of age, but there was no association between FR at 6 years of age and later BMI. Thus, they could not support the notion that eating behaviour predicts future BMI.

As previously noted, meat consumption at 12 months was significantly higher in the intervention group whilst FR at 6 years of age was significantly lower. It is, therefore, possible that the intervention indirectly influenced childhood BMI via these pathways. However, a larger sample is needed to explore this further. Interestingly, although there are concerns that a higher protein intake from animal food sources might promote unhealthy weight gain, previous studies have reported that meat and red meat intake were not related to an increase in a prevalence of being overweight and obese [27,28,29].

At 6 years of age, children in the NCFG group had significantly lower HAZ; however, it was predicted by LAZ at 6 months of age and there was no significant effect of the intervention on a change in HAZ from 12 months to 6 years of age. Multivariate analyses showed that HAZ at 6 years of age was positively predicted only by LAZ at 6 months, highlighting the importance of linear growth in early infancy.

The main strength of this study was that participants participated in an RCT with detailed dietary intake and anthropometric data collected in infancy from 6 to 12 months of age. The main limitations were the small sample size, attrition (58.8% of the original trial participants studied) and a lack of data on physical activity, which is an important factor influencing child weight status.

## 6. Conclusions

These findings from a population of children living under socioeconomic constraints suggest that infants randomised to an intervention promoting breastfeeding alongside appropriate CF had lower food responsiveness in mid-childhood. Childhood BMI was not directly influenced by the intervention, but the results were consistent with an indirect effect of the intervention on childhood BMI via effects from meat consumption in infancy and childhood FR. However, further longitudinal studies with a larger sample size are needed to explore this further.

## Figures and Tables

**Figure 1 nutrients-16-02311-f001:**
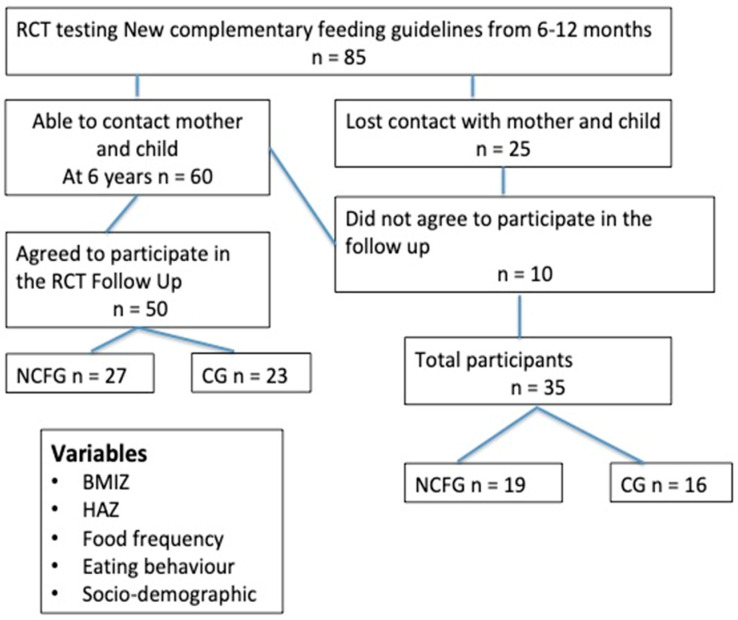
Flowchart showing progress of participants in a RCT testing the effects of new complementary feeding guidelines on eating behavior, food composition and growth in Colombian infants with 6 year follow-up.

**Table 1 nutrients-16-02311-t001:** Characteristics at recruitment (6 months) of the 50 infants followed up at 6 years of age and 35 infants not followed up by randomised group.

Variable	Followed Up at 6 Years of Age (n = 50)			Not Followed Up (n = 35)		
	NCFG Group (n = 23)	CG (n = 27)	Total n = 50	NCFG Group (n = 19)	CG (n = 16)	Total (n = 35)
Child’s age (mo)	6.6 (±0.3)	6.5 (±0.2)	6.5 (±0.3)	6.3 (±0.2)	6.3 (±0.3)	6.3 (±0.2)
Sex						
Male, n (%)	12 (52.2)	13 (48.1)	25 (50)	8 (42.1)	8 (50)	16 (45.7)
Female, n (%)	11 (47.8)	14 (51.8)	25 (50)	11 (57.9)	8 (50)	19 (54.3)
Gestational age (wks)						
Mean (SD)	39.0 (±0.9)	39.2 (±0.9)	39.2 (±0.9)	39.1 (±1.2)	39.1 (±1.0)	39.1 (±1.1)
Birth weight (g), mean (SD)	2980 (±264)	3134 (±379)	3063 (±337)	3133 (±286)	3133 (±401)	3133 (±338)
Birth length, mean (SD)	49.2 (±1.7)	50.4 (±1.9)	49.8 (±1.8)	50.3 (±1.4)	50.2 (±1.9)	50.3 (±1.6)
CF < 6 mo, n (%)	10 (43.5)	12 (44.4)	21 (42.0)	9 (47.4)	9 (56.3)	18 (51.4)
CF > 6 mo, n (%)	13 (56.5)	15 (55.6)	29 (58)	10 (52.6)	7 (43.2)	17 (48.6)
LAZ at 6 mo, mean (SD)	−1.11 (±1.0)	−0.71 (±1.0)	−0.89 (±1.1)	−0.68 (±1.2)	−0.65 (±0.8)	−0.66 (±1.0)
WLZ at 6 mo, mean (SD)	0.22 (±0.9)	0.83 (±1.0)	0.55 (±1.0)	−0.05 (±0.7)	0.4 (±1.1)	0.15 (±0.9)
Mother’s age (y)						
Mean (SD)	32.4 (±7.3)	32.7 (±9.7)	32.6 (±8.7)	23.5 (±6.8)	22.1 (±6.5)	22.9 (±6.6)
Marital status, n (%)						
Single	6 (26.1)	4 (14.8)	10 (20)	5 (26.3)	4 (25)	9 (25.7)
Married	4 (17.4)	5 (18.5)	9 (18)	2 (10.5)	0 (0)	2 (5.7)
Living together	12 (52.2)	15 (55.6)	27 (54)	12 (63.2)	12 (75)	24 (68.6)
Mother’s education, n (%)						
Primary	1 (4.3)	1 (3.7)	2 (4)	0 (0)	2 (13.3)	2 (5.9)
Mid-high school	4 (17.4)	2 (7.4)	6 (12)	8 (42,1)	5 (33.3)	13 (38.2)
Completed high school	12 (52.2)	12 (44.4)	24 (48)	7 (36.8)	7 (46.7)	14 (41.2)
Technician	3 (33.3)	9 (33.3)	12 (24)	3 (15.8)	1 (6.7)	4 (11.8)
Undergraduate	1 (4.3)	1 (3.7)	2 (4)	0 (0)	0 (0)	0 (0)
Family size, mean (SD)	5.1 (±1.1)	5.8 (±1.7)	5.5 (±1.5)	5.5 (±1.5)	4.8 (±2.9)	5.1 (±2.2)
House owner, n (%)						
Yes	7 (30.4)	5 (18.5)	3 (6)	0 (0)	0 (0)	0 (0)
No	16 (69.6)	22 (81.5)	47 (94)	19 (100)	16 (100)	35 (100)

NCFG: new complementary feeding guideline; CG: control group (current complementary feeding); mo: months; wks: weeks; y: years; CF: complementary feeding; LAZ: length for age Z-score; WLZ: weight for length Z-score; n: sample size; SD: standard deviation.

**Table 2 nutrients-16-02311-t002:** Characteristics of the 50 infants followed up at 6 years of age according to randomised group.

Variable	NCFG Group (n = 23)	CG (n = 27)	*p*-Value
Child’s age (years), mean (SD)	6.6 (±0.3)	6.5 (±0.2)	0.27
Sex			0.50
Male, n (%)	12 (52.2)	13 (48.1)	
Female, n (%)	11 (47.8)	14 (51.8)	
Months of EBF, mean (SD)	5.6 (±0.6)	5.3 (±0.8)	0.22
Months of TBF, mean (SD)	25.4 (±12.9)	20.2 (±8.8)	0.10
CF < 6 mo, n (%)	10 (43.5)	12 (44.4)	0.58
CF > 6 mo, n (%)	13 (56.5)	15 (53.6)	
LAZ at birth	−1.9 (±0.9)	0.43 (±0.99)	**0.03**
LAZ at 12 mo, mean (SD)	−1.3 (±1.0)	−0.99 (±1.0)	0.27
WLZ at 12 mo, mean (SD)	0.00 (±1.0)	0.57 (±0.9)	**0.04**
BMIZ at 12 mo, mean (SD)	0.17 (±0.97)	0.72 (±0.9)	**0.05**
Mother’s stature (cm), mean (SD)	154.8 (±7.4)	158.3 (±5.4)	0.08
Mother’s BMI, mean (SD)	24.9 (±3.0)	25.5 (±4.7)	0.61
Socioeconomical status, n (%)			0.49
Level 1	1 (4.3)	0 (0)	
Level 2	16 (69.6)	18 (66.7)	
Level 3	6 (26.0)	9 (33.3)	

NCFG: new complementary feeding guideline; CG: control group (current complementary feeding); mo: months; n: sample size; SD: standard deviation; EBF: exclusive breastfeeding; TBF: total breastfeeding; CF: complementary feeding; LAZ: length for age Z-score; WLZ: weight for length Z-score; BMIZ: body mass index Z-score; Bold text indicates *p*-value (test probability) significance ≤ 0.05.

**Table 3 nutrients-16-02311-t003:** Child growth, food consumption and eating behaviour at 6 years of age according to randomised group.

Variable	NCFG Group (n = 23)	CG (n = 27)	*p*-Value
** *Growth (mean (SD))* **			
BMI	15.7 (±1.3)	16.2 (±1.7)	0.23
BMIZ	0.13 (±0.8)	0.46 (±1.0)	0.21
HAZ	−0.74 (±0.8)	−0.14 (±0.9)	**0.02**
Change in BMIZ from 6 mo to 6 y	0.08 (±1.2)	−0.33 (±1.2)	0.29
Change in BMIZ from 12 mo to 6 y	0.003 (±1.3)	−0.24 (±1.2)	0.55
Change in HAZ from 6 mo to 6 y	0.28 (±0.8)	0.49 (±0.8)	0.36
Change in HAZ from 12 mo to 6 y	0.48 (±0.8)	0.77 (±0.9)	0.24
Overweight, n (%)	3 (13)	5 (18.5)	0.54
Obese, n (%)	0 (0)	1 (3.7)	
** *Mother’s weekly (number of times per week) food consumption (mean (SD))* **			
Red meat	4.55 (±3.6)	3.33 (±2.1)	0.91
Milk	7.45 (±6.4)	7.38 (±5.8)	0.97
Vegetables	12.0 (±9.8)	8.0 (±6.2)	0.12
Fruit	11.8 (±6.0)	11.6 (±14.6)	0.95
Processed food	0.26 (±0.45)	1.24 (±1.7)	**0.** **019**
Soft drinks	1.4 (±3.2)	3.4 (±4.9)	**0.** **049**
Sugary drinks	0.21 (±0.5)	1.05 (±1.7)	**0.** **045**
Chocolate drinks	2.89 (±2.8)	2.81 (±3.0)	0.92
** *Child’s weekly (number of times per week) food consumption (mean (SD))* **			
Red meat	3.5 (±1.8)	3.5 (±2.0)	0.99
Milk	10.2 (±4.3)	8.1 (±5.0)	0.11
Vegetables	8.7 (±4.7)	8.1 (±4.2)	0.64
Fruit	14.9 (±5.3)	13.5 (5.4)	0.35
Processed food	3.0 (±2.6)	2.9 (±2.4)	0.78
Soft drinks	3.0 (±4.3)	2.1 (±1.8)	0.31
Sugary drinks	1.9 (±3.1)	1.3 (±1.6)	0.41
Chocolate drinks	3.5 (±2.4)	5.9 (±5.1)	**0.048**
** *Eating Behaviour (mean (SD))* **			
Food responsiveness (FR)	2.45 (±1.1)	3.4 (±0.9)	**0.001**
Emotional overeating (EOE)	2.6 (±0.7)	2.8 (±0.9)	0.30
Enjoyment of food (EF)	4.4 (±0.7)	4.3 (±0.8)	0.63
Desire to drink (DD)	4.5 (±0.8)	4.6 (±0.6)	0.64
Satiety responsiveness (SR)	3.5 (±0.8)	3.1 (±0.9)	0.20
Slowness in eating (SE)	3.4 (±0.9)	3.2 (±0.8)	0.42
Emotional undereating (EUE)	3.40 (±1.1)	3.0 (±1.2)	0.23
Food fussiness (FF)	3.5 (±0.6)	3.7 (±0.5)	0.17

NCFG: new complementary feeding guideline; CG: control group (current complementary feeding); mo: months; y: years; n: sample size; BMIZ: body mass index Z-score; HAZ: height for age Z-score; SD: standard deviation; Bold text indicates *p*-value (test probability) significance ≤ 0.05.

**Table 4 nutrients-16-02311-t004:** Correlations between child’s and mother’s daily and weekly food frequency consumption.

Food Consumption	Correlations Between Child’s and Mother’s Daily Food Consumption *	Correlations Between Child’s and Mother’s Weekly Food Consumption *
Milk		
n	41	41
r	0.539	0.341
*p*	0.0001	0.029
Milk dairy		
n	41	41
r	0.417	0.413
*p*	0.007	0.007
Vegetables		
n	40	40
r	0.528	0.465
*p*	0.0001	0.003
Fruit		
n	40	40
r	0.398	0.408
*p*	0.011	0.009
Red meat		
n	41	41
r	0.339	0.599
*p*	0.030	0.0001

* Spearman correlation analyses.

**Table 5 nutrients-16-02311-t005:** Predictors of BMIZ and HAZ at 6 years of age.

Variable	Standardised β Coefficient	95% CI	*p*-Value
**BMIZ at 6 years of age**			
Model 1			
Mother’s BMI	0.084	0.011 to 0.158	0.025
Model 2			
FR	0.293	0.064 to 0.523	0.014
Weekly red meat consumption at 12 mo	−0.169	−0.310 to −0.028	0.020
**HAZ at 6 years of age**			
Model 1			
LAZ at 6 mo	0.656	0.468 to 0.844	0.0001
Model 2			
LAZ at 6 mo	0.683	0.491 to 0.874	0.0001

BMIZ: body mass index Z-score; BMI: body mass index; FR: food responsiveness; HAZ: height Z-score; mo: months; LAZ: length Z-score. *BMIZ model 1*: stepwise linear regression model with BMIZ at 6 years of age as dependent variable and baseline variables of birth weight, child’s sex, mother’s educational level, mother’s occupation, mother’s BMI and intervention group (GC-NCFG) as independent variables. *BMIZ model 2:* stepwise linear regression model with BMIZ at 6 years of age as dependent and post-randomisation variables (FR and weekly red meat consumption at 12 months as number of times). *HAZ model 1:* stepwise linear regression model with HAZ at 6 years of age as dependent variable and baseline variables of mother’s height, birth length, sex, number of children, mother’s educational level, mother’s occupation, HAZ at 6 months and intervention group (CG-NCFG). *HAZ model 2:* stepwise linear regression model with HAZ at 6 years of age as dependent variable and post-randomisation of WLZ at 12 months as independent variable. Only significant variables for BMZ and HAZ at 6 years of age are shown in the table.

## Data Availability

The data presented in this study may be available on request from the corresponding author pending permission from the ethics committee.

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
