# Peer review of "Effects of New Complementary Feeding Guidelines on Eating Behaviour, Food Consumption and Growth in Colombian Children: 6-Year Follow-Up of a Randomised Controlled Trial"

_nutrients, 2024, doi:10.3390/nu16142311_

Round 1

Reviewer 1 Report

Comments and Suggestions for Authors

This paper assesses the longer-term impact of new complementary feeding guidelines in Bogota, Colombia on the growth, good consumption, and eating behavior of children at 6 years of age. While I appreciate the spirit of the study, the authors’ main conclusions are not supported by the results of their analyses, and the sample size is extremely limited (50 children). These, and other comments, are listed below.

Abstract: The setting of the study/RCT is not mentioned anywhere in the abstract and should be added somewhere (e.g., “Our previous RCT reported that new CF guidelines (NCFG) implement in …”)

Line 10: It would be helpful if you specified the age of the children at this first, primary assessment (vs 6 years of age at the follow-up).

Line 16: I question whether it is appropriate to highlight results that are not statistically significant in this way, especially since they seem a bit cherry-picked (e.g., there is also a “trend toward” more frequent soft drink consumption in the NFCG, but you chose not to report that, here and in the main paper), especially given the extremely small sample size. I suggest you only report statistically significant results in the abstract and note that difference in the frequency of consumption of other food groups were not statistically different between the two groups. You also do not report that HAZ was significantly lower in the NFCG at 6 years. Again, it seems inappropriate to cherry-pick which results are and are not reported (fine to also note that change in HAZ was not significant, but I think you do need to report this result in the abstract and also in the discussion section).

Lines 21-22: I’m confused here, as the results showed no difference in BMIZ or the change in BMIZ between the two groups. If you are referring to the multivariate regression results, please see my comments below about the need to exclude post-randomization variables if you are trying to estimate the effect of treatment group on growth or food consumption.

Lines 27-29: Please add references for this statement.

Lines 46-59: Here, too, the setting/location of the study is not mentioned and should be added.

Line 66: Can you add a little more information about how “socioeconomic constraints” were defined? This is helpful contextual information.

Line 88: Not clear what is meant by HAZ/age, height for age z-score for age?

Lines 100-102: Iy would be helpful if you added a bit more information about some of these scales. For example, what does food responsiveness measure? What about desire to drink? Etc.

Lines 113 and 115: Should rather be HAZ, right, given the age of the children?

Lines 131-132: You say “but children in the CG had significantly higher LAZ at birth and WLZ at 12mo (Table 2)…” but I don’t see LAZ at birth reported anywhere in Table 2.

Table 1. Not clear what the variable “House´s owner” means.

Table 2. Please describe how the SES variable was constructed and what the levels correspond to.

Lines 149-152/Table 3: I don’t see the proportion of overweight and obesity reported in Table 3 and I think it should be added.

Table 3: Your use of bolding to indicate statistical significance is inconsistent (0.05 is bolded for chocolate drinks for children but no sugary drinks for mothers). Also, please show at least two decimal places for all p-values. Also, the asterisk after food responsiveness does not seem to correspond to anything in the Table notes (3.4 (±0.9)*). Also, please clarify what the food consumption numbers mean (frequency in the past week, presumably).

Lines 161-162: Also a trend towards more frequent soft drink consumption, right? Important to not pick and choose which results you highlight.

Sections 4.4 and 4.5: The results of the multiple linear regressions need to be presented in table form, not just described in the text. Also, I’m unclear about what you are trying to do here. If you are trying to assess the impact of the intervention on BMIZ, HAZ, and eating behavior in an adjusted model, then you would definitely need to only include a dummy for group and BASELINE covariates (i.e., covariates collected pre-randomization, so anything measured after randomization that could have been influence by treatment group needs to be excluded). To make conclusions that imply causality like (from the abstract) “These findings, are consistent with positive effects of the intervention on child BMI at 6 years…” and (from the conclusions section) “These findings, from a population of children living under socio- economic constraints, are consistent with positive effects of an intervention promoting breastfeeding alongside appropriate CF, on child BMI at 6 years…” when in fact the difference in BMI, BMIZ, and the change in the two were not statistically different in the unadjusted models, you definitely need to establish the significance of group on BMI of BMIZ in an adjusted model that does not include any post-randomization variables (for example food responsiveness). If you also want to look at some kind of pathways analysis as a secondary analysis if group is significant in the adjusted model, then that can be done separately and needs to be well-explained in the methods section. But as it is, the implication that the intervention had an impact of BMI/BMIZ at 6 years of age is not supported by the findings.  

Lines 213-215: I don’t see these result reported anywhere in the paper. If you are going to make conclusions like this (which I find to be a serious stretch), you certainly at least need to present the quantitative results.

Lines 260-261: Again, this result is not reported anywhere (unless I completely missed it). You discuss this result multiple times – if you want to discuss it, the result actually needs to be presented (n and proportion overweight and obese by group).

Comments on the Quality of English Language

The quality of English Language is generally adequate (a typo here and there, etc.)

Author Response

Dear reviewer 1

Thank you so much

Reviewer 2 Report

Comments and Suggestions for Authors

This is an interesting study. However, some methodological concerns are needed to address.

Method

Line 63- What are eligible study participants? How and where were study participants recruited? How did the author assess the adherence to the NCFG?

Line 93- Are the FFQ validated?

Line 99- Is the Spanish-version CEBQ back-translated and validated?

Line 105- Did the authors perform analyses of intention-to-treat?

Table 2

Abbreviations (for example, EBF and TBF) should be spelled out.

Author Response

Dear reviwer 1.

Thank you so much 

Round 2

Reviewer 1 Report

Comments and Suggestions for Authors

The authors have done a nice job addressing most of my previous comments, although a few issues still remain. They are listed below.

Lines 71-72: Ok, but what were the socioeconomic strata based on? An asset index? Household expenditures? Please specify.

Line 75: Typo. Tape water should be tap water.

Lines 278-279: There is only one obese child in the entire sample; this certainly cannot be characterized as a trend toward higher rates of obesity in the CG.

Table 5. This table is very unclear as presented. Please revise the table to make it interpretable as a stand-alone table.

Lines 533, 534: Typos. Pos-randomisation instead of post-randomization; length z-score instead of length for age z-score.

Comments on the Quality of English Language

The paper would benefit from a careful proofreading for typos, missing words, etc. 

Author Response

Dear Editor

Thank you for considering our manuscript “Effects of new complementary feeding guidelines on eating behavior, food consumption and growth in Colombian infants: 6 year follow-up of a RCT” for publication in the Nutrients Journal.

We have provided responses to reviewer 1’s comments below and have highlighted the changes in the text.

Reply to the Review Report

Manuscript Title: Effects of new complementary feeding guidelines on eating behaviour, food consumption and growth in Colombian infants: 6-year follow-up of a RCT

Manuscript ID nutrients-2990200

Review Report 1

 Lines 71-72: Ok, but what were the socioeconomic strata based on? An asset index? Household expenditures? Please specify.

Yours faithfully,

Gilma Olaya

Mary Fewtrell